# Denoise while Aggregating: Collaborative Learning in Open-Domain Question Answering

## Abstract

The open-domain question answering (OpenQA) task aims to extract answers that match specific questions from a distantly supervised corpus. Unlike supervised reading comprehension (RC) datasets where questions are designed for particular paragraphs, background sentences in OpenQA datasets are more prone to noise. We observe that most existing OpenQA approaches are vulnerable to noise since they simply regard those sentences that contain the answer span as ground truths and ignore the plausible correlation between the sentences and the question. To address this deficiency, we introduce a unified and collaborative model that leverages alignment information from query-sentence pairs in a small-scale supervised RC dataset and aggregates relevant evidence from distantly supervised corpus to answer open-domain questions. We evaluate our model on several real-world OpenQA datasets, and experimental results show that our collaborative learning methods outperform the existing baselines significantly.

## 1 Introduction

Driven by neural networks with attention mechanisms and large-scale supervised reading comprehension (RC) datasets, teaching the machine to read within close-domain paragraphs has grown in leaps and bounds (Chen et al., 2016; Dhingra et al., 2017a; Cui et al., 2017; Shen et al., 2017; Wang et al., 2017). Recently, a new challenge called open-domain question answering (OpenQA) arises with the publishment of several distantly supervised OpenQA datasets. Different from supervised RC datasets where designed questions are oriented to particular paragraphs, background sentences in OpenQA datasets are usually retrieved by information retrieval systems using simple matching tactics, which leads to the noise-prone nature of this task.

Existing approaches for OpenQA can be roughly divided into the aggregation-based methods (Wang et al., 2018b; Lin et al., 2016; Clark & Gardner, 2018) that aggregate all possible evidence to predict the answer and the selection-based methods (Choi et al., 2017; Wang et al., 2018a; Min et al., 2018) that only select a minimal set of sentences for question answering. These models simply follow the distantly supervised assumption, i.e., regarding all the sentences that contain the answer span as ground truths. However, in many cases, plausible correspondence may not exist between the question and those sentences. For example, given the question "*Which country has the fourth largest population?*", the assumption of distant supervision regards both of the two retrieved sentences to be valid: (1) "*With well over 210 million people, Indonesia is the fourth most populous country in the world.*" (2) "*..., Indonesia is New Zealand's fourth largest source of imports.*" In fact, instead of treating those candidate sentences equally, it makes more sense to give different semantic labels to those sentences based on their plausible correlation with the question. For the above example, the first sentence is supposed to have a higher semantic label than the second since it precisely answers the given question while the second fails.

In this paper, we propose an aggregation-based OpenQA model consisting of a sentence discriminator which explicitly computes the query-sentence matching score and a sentence reader that selects a proper answer span from the given sentence. We show that by explicitly mearsuring the relatedness between the sentence and the question, the model could effectively leverage structure information at the sentence level to promote the overall performance of question answering.

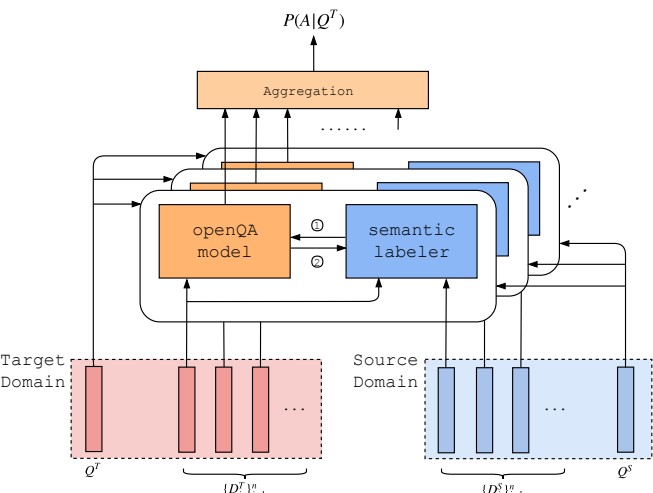

Figure 1: The high level architecture of our model consists of an aggregation-based OpenQA model and a semantic labeler. Our denoising strategies involve: (1) incoporating source domain knowledge to denoise the OpenQA task; (2) sharing information between two domains via collaborative learning.

Moreover, to address the problem of lacking supervised signal in distantly supervised OpenQA datasets, we learn a semantic labeler by incorporating the query-sentence alignment information from a small-scale set of supervised reading comprehension. We propose two denoising strategies including semi-supervised learning with semantic labels and collaborative learning that further encourages cross-domain information sharing.

We evaluate our work on three large-scale OpenQA datasets including Quasart-T (Dhingra et al., 2017b), SearchQA (Dunn et al., 2017) and the unfiltered version of TriviaQA (Joshi et al., 2017). The experiment results show that our model achieves significant improvement as compared to the published baselines on these datasets. Besides, we show that via collaboratively learning with the semantic labeler, our sentence discriminator is capable of distinguishing relevant sentences from those irrelevant ones, and significantly promoting the answer prediction process in the overall pipeline.

## 2 METHODOLOGY

### 2.1 TASK FORMULATION

The OpenQA task aims to extract the answer of a given question from a set of distantly supervised texts. Formally, given a question with $m$ words $Q = \{q_1, q_2 \ldots, q_m\}$ and a set of $n$ distantly supervised background sentences $C = \{D_1, \cdots, D_i, \cdots, D_n\}$, the task is to extract an answer span from the background sentence set $C$.

### 2.2 MODEL ARCHITECTURE

As illustrated in Figure 1, the high-level structure of our model is comprised of a semantic labeler and an aggregation-based OpenQA model.

The aggregation-based OpenQA model handles the question-answering task on the distant-supervised dataset. During training, we maximize the probability of the answer span $A$ being correct given the question $Q$ by aggregating probabilities in various paragraphs. The semantic labeler is learned from the supervised reading comprehension dataset $\mathcal{S} = \{(Q^{\mathcal{S}}, C^{\mathcal{S}}, A^{\mathcal{S}}, y^{\mathcal{S}}), \ldots\}$ to capture valid query-sentence correlation. Then, it is used to produce soft labels $\hat{y}^{\mathcal{T}}$ for query-sentence pairs in the distant-supervised dataset $\mathcal{T} = \{(Q^{\mathcal{T}}, C^{\mathcal{T}}, A^{\mathcal{T}}), \ldots\}$ where accurate sentence labels are absent.

In the following sections, we first describe our OpenQA model in details and then demonstrate our two denoising strategies.

## 2.3 AGGREGATION-BASED OPENQA MODEL

Our aggregation-based OpenQA model consists of a sentence discriminator and a sentence reader. For both of the two sub-models, we use a generic encoder to construct representations for the question and the sentence. The decoders of the two sub-models, however, are more task-specific.

### 2.3.1 SENTENCE DISCRIMINATOR

The sentence discriminator is divided into the encoder part and the sentence-level decoder part, which will be illustrated in the following respectively.

**Encoder**

The encoder takes a sentence and question as input and yields the respective representations. Without loss of generality, we adopt an encoder similar to Chen et al. (2017) in structure which involves an embedding layer, a context-query interactive layer and an encoder layer. We denote the word embeddings as $\boldsymbol{D}^{emb} \in \mathbb{R}^{d \times l}$ for the sentence and $\boldsymbol{Q}^{emb} \in \mathbb{R}^{d \times m}$ for the question. First, the embeddings are fed into the context-query interactive layer to obtain the interactive embedding of the sentence:

$$\boldsymbol{S} = f(\boldsymbol{D}^{emb})^T f(\boldsymbol{Q}^{emb}) \in \mathbb{R}^{l \times m}, \quad \boldsymbol{A}_{i:} = \frac{\exp(\boldsymbol{S}_{i:})}{\sum_{k=1}^{l}, \exp(\boldsymbol{S}_{k:})} \in \mathbb{R}^m, \quad \boldsymbol{D}^q = \boldsymbol{Q}^{emb} \boldsymbol{A}^T \in \mathbb{R}^{h \times l}, \quad (1)$$

where we use ReLU as our nonlinear function $f(\cdot)$. Then, we encode the embeddings using bidirectional LSTM to capture context-aware representations.

$$\boldsymbol{D}^{enc} = \text{BiLSTM}([\boldsymbol{D}^{emb}, \boldsymbol{D}^q]), \quad \boldsymbol{Q}^{enc} = \text{BiLSTM}(\boldsymbol{Q}^{emb}). \quad (2)$$

**Sentence-level Decoder**

The sentence-level decoder computes a score for the sentence given $\boldsymbol{D}^{enc}$ and $\boldsymbol{Q}^{enc}$. We first merge the encoded question representation to obtain a fix length hidden vector with self-attention weights:

$$\boldsymbol{\alpha} = \text{softmax}(\boldsymbol{w}_1^T \boldsymbol{Q}^{enc}) \in \mathbb{R}^m, \quad \tilde{\boldsymbol{q}}_{enc} = \sum_{i=1}^{m} (\boldsymbol{\alpha}_i \boldsymbol{Q}_{i:}^{enc}) \in \mathbb{R}^h, \quad (3)$$

where $\boldsymbol{w}_1 \in \mathbb{R}^h$ is a trainable weight vector. Next, we calculate bilinear similarity to construct a question-aligned sentence representation $\boldsymbol{D}^{align}$, which is further used to calculate the sentence score.

$$\boldsymbol{D}^{align} = \tilde{\boldsymbol{q}}_{enc} \boldsymbol{W}_2 \boldsymbol{D}^{enc} \in \mathbb{R}^{h' \times l}, \quad (4)$$

$$\text{score} = \text{softmax}(\boldsymbol{W}_3 \max_i(\boldsymbol{D}_{:i}^{align})). \quad (5)$$

where $\boldsymbol{W}_2 \in \mathbb{R}^{h \times h' \times h}, \boldsymbol{W}_3 \in \mathbb{R}^{2 \times h'}$ are trainable parameters. Note that $h'$ is a hyperparameter varied between different models.

### 2.3.2 SENTENCE READER

The sentence reader consists of the same encoder as the sentence discriminator and a token-level decoder that computes the probability of the given text span being the answer.

**Token-level Decoder**

First, we merge question representations using (3) and obtain $\tilde{\boldsymbol{q}}_{enc}$, which is later used to compute the probabilities of each token being the two ends of the span as given by (6). The answer probability is obtained by multiplying the probabilities for the start token $D_s$ and the end token $D_e$.

$$P_s(D_j) = \text{softmax}(\tilde{\boldsymbol{q}}_{enc} \boldsymbol{W}_4 \boldsymbol{D}_{:j}^{enc}), \quad P_e(D_j) = \text{softmax}(\tilde{\boldsymbol{q}}_{enc} \boldsymbol{W}_5 \boldsymbol{D}_{:j}^{enc}), \quad (6)$$

$$P(A|D, Q) = P_s(D_s) \times P_e(D_e). \quad (7)$$

Here, $\boldsymbol{W}_4 \in \mathbb{R}^{h \times h}, \boldsymbol{W}_5 \in \mathbb{R}^{h \times h}$ are learnable parameters during training.

### 2.3.3 TRAINING AND PREDICTION

We train the OpenQA model in two stages. Firstly, the sentence discriminator is trained on the OpenQA dataset to produce a score $R_i$ for each sentence in the set $\{D_i\}_{i=1}^n$. We give each sentence a distantly supervised label $\tilde{y} = \{0, 1\}$ indicating whether the sentence contains a text span that matches the answer exactly and optimize the cross-entropy loss.

$$\mathcal{L}_D = \frac{1}{n} \sum_{i=1}^n -\tilde{y}_i \log(R_i) - (1 - \tilde{y}_i) \log(1 - R_i). \tag{8}$$

Then, in the second stage, we fix the sentence disctiminator and train the sentence reader to optimize the overall probability of the answer $P(A|Q)$. We normalize each sentence score $R_i$ to approximate the distribution $P(D_i|Q)$, which is formally calculated as:

$$P(D_i|Q) = \frac{R_i}{\sum_{j=1}^n R_j}, \quad P(A|Q) = \sum_{i=1}^n P(A|D_i, Q)P(D_i|Q). \tag{9}$$

And the overall loss of the aggregation-based OpenQA model is defined as the negative sum of log probabilities:

$$\mathcal{L}_{QA} = -\frac{1}{|\mathcal{T}|} \sum_{(A,Q)\in\mathcal{T}} \log P(A|Q). \tag{10}$$

When predicting the answer, we first decode top $k$ answer spans $\{A_j\}_{j=1}^k$ for each sentence $D_i$ based on $P(A_j|D_i, Q)$ and build a candidate set with $k \times n$ candidates. Then, for each candidate, we calculate the overall probability $P(A_j|Q)$ by considering its total occurrences in the background sentences given by (9). Finally, we select the candidate with the maximimal overall probability.

### 2.4 DENOISING STRATEGIES ON OPENQA

Our denoising strategies on the OpenQA task combine the aggregated OpenQA model and the semantic labeler. In the following, we first introduce our semantic labeler and then two denoising models respectively.

### 2.4.1 SEMANTIC LABELER

The semantic labeler aims to transfer knowledge from the supervised dataset $\mathcal{S} = \{(Q^{\mathcal{S}}, D^{\mathcal{S}}, A^{\mathcal{S}}, y^{\mathcal{S}}), \dots\}$ which we refer to as the source domain, to the distantly supervised datasets $\mathcal{T} = \{(Q^{\mathcal{T}}, D^{\mathcal{T}}, A^{\mathcal{T}}), \dots\}$ which we refer to as target domains. To this end, we employ a structure similar to the sentence discriminator. For multi-task requirement, we construct the embedding layer with both domain's vocabularies to encourage the model to learn unified representations for words that lie within the intersection of both domain's vocabularies. As for the decoder, we use the sentence-level decoder with $h' = 1$ for better scalability on the distant-supervised dataset. This setting improves the generalization of the semantic labeler and prevents it from overfitting on the small-scale source domain during training.

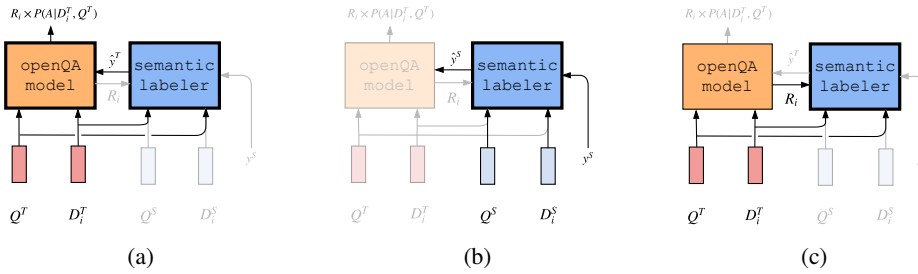

Figure 2: The training procedure of our denoising schemes. (a) Train the unified OpenQA model with the semantic labeler. (b) Train the semantic labeler with supervised labels. (c) Marginalize the semantic labels on target domain. Faded parts of the model are disabled and models with bold frames are updated in the corresponding stages.

### 2.4.2 DENOISING MODELS

We propose two denoising models using the semantic labeler including semi-supervised learning with semantic labels and collaborative learning with semantic labels.

**SSL** (Semi-supervised Learning with Semantic Labels)

In SSL, the aggregation-based OpenQA model directly learns from the semantic labels computed by the semantic labeler which is pre-trained on the source domain $\mathcal{S}$. First, given $n'$ sentences and a corresponding question from the source domain, the semantic labeler learns to compute soft labels $\hat{y}^{\mathcal{S}}$ by optimizing the following loss function.

$$\mathcal{L}_{SL} = \frac{1}{n'} \sum_{i=1}^{n'} -y_i^{\mathcal{S}} \log(\hat{y}_i^{\mathcal{S}}) - (1 - y_i^{\mathcal{S}}) \log(1 - \hat{y}_i^{\mathcal{S}}). \quad (11)$$

Then, on the target domain $\mathcal{T}$, we train the aggregated-based OpenQA model with semantic labels $\hat{y}^{\mathcal{T}}$ generated by the semantic labeler. For regularization purpose, we add a weighted discriminator loss to the original reader loss:

$$\mathcal{L}_{WD} = \frac{1}{n} \sum_{i=1}^{n} -\hat{y}_i^{\mathcal{T}} \log(R_i^{\mathcal{T}}) - (1 - \hat{y}_i^{\mathcal{T}}) \log(1 - R_i^{\mathcal{T}}). \quad (12)$$

$$\mathcal{L}_{SSL} = \mathcal{L}_{QA} + \alpha \mathcal{L}_{WD}, \quad (13)$$

where $\alpha$ is a hyperparameter determining the scale of the regularization by the discriminator. To keep the united model aware of the global contexts, we share the embedding layer of the two models here.

**CSL** (Collaborative Learning with Semantic Labels)

To reduce the gap between the source $\mathcal{S}$ and target domain $\mathcal{T}$, we propose a collaborative model in which the semantic labeler is trained in a multitask fashion to encourage cross-domain information sharing. In this model, the semantic labeler is simultaneously trained on the source domain $\mathcal{S}$ to leverage supervised information and the target domain $\mathcal{T}$ to produce marginalized semantic labels $\hat{y}^{\mathcal{T}}$.

$$\mathcal{L}_{TL} = \frac{1}{n} \sum_{i=1}^{n} -R_i \log(\hat{y}_i^{\mathcal{T}}) - (1 - R_i) \log(1 - \hat{y}_i^{\mathcal{T}}) \quad (14)$$

$$\mathcal{L}_{CSL} = \mathcal{L}_{TL} + \beta \mathcal{L}_{SL}, \quad (15)$$

where $\beta$ is a hyperparameter determining the intensity of the supervised signal obtained from the source domain $\mathcal{S}$.

### 2.4.3 TRAINING AND PREDICTION

Figure 2 shows the training procedure. For the semi-supervised model, we first train the semantic labeler on the source domain and then train the OpenQA model by optimizing $\mathcal{L}_{SSL}$ with semantic labels as shown in 2(b) and 2(a). In the collaborative model, the semantic labeler is jointly trained with the OpenQA model. As shown in 2(b) and 2(c), we iteratively forward data from two domains to optimize $\mathcal{L}_{CSL}$ while fixing the parameters of the OpenQA model. Then in the same iteration, we train the OpenQA model as shown in 2(a). Note that in the two models above, the OpenQA model is trained as a whole without fixing any parts. The prediction process is identical to the raw OpenQA model as mentioned in Section 2.3.3.

## 3 EXPERIMENTS

### 3.1 DATASETS AND EVALUATION METRICS

**SQuAD** The Stanford Question Answering Dataset (SQuAD) (Rajpurkar et al., 2016) is a supervised reading comprehension dataset constructed on Wikipedia. We split the paragraph into sentences and use it as our source domain.

**Quasart-T** (Dhingra et al., 2017b) consists of 43,000 factoid question-answer pairs accompanied by a large-scale background corpus. We train and evaluate our model on the short version of Quasart-T.

**SearchQA** (Dunn et al., 2017) is a large scale open QA dataset containing 140,000 question-answer pairs crawled from J! Archive. Web page snippets retrieved by Google are used as the background corpus.

**TriviaQA** (Joshi et al., 2017) is a distantly supervised dataset for reading comprehension consisting of trivia question-answer pairs. We use the unfiltered version of TriviaQA in our experiment.

The statistics of these datasets are shown in Table 3.1. We use ExactMatch (EM) and F1 score to evaluate the overall performance on the OpenQA task. While the exact match is a binary score denoting whether part of the sentence matches to the answer exactly, the F1 score computes the mean of precision and recall at the token level. As for the evaluation of our sentence discriminator, we compute accuracy of the top-1 ranked sentences based on their ranking scores.

| Dataset | Train | Dev | #ACS |
|---------|-------|-----|------|
| SQuAD* | 87,599 | 10,570 | 5.06 |
| Quasar-T | 28,485 | 2,277 | 100 |
| SearchQA | 99,811 | 13,893 | 50 |
| TriviaQA | 66,828 | 11,313 | 100 |

Table 1: Statistics of the datasets used in this paper. #ACS: average number of background sentences per example. *: We split each paragraph into sentences on SQuAD.

## 3.2 EXPERIMENTAL SETUP

To process the corpus, we use the script provided by Wang et al. (2018a). We also perform a grid search and evaluate our model on the development sets to choose the optimal hyperparameters. For word embeddings, we use 300-dimensional GloVe (Pennington et al., 2014) word vectors. We set the hyperparameters of our encoder following Chen et al. (2017). For the decoder, we set $h' = 1$ for the semantic labeler and $h' = 64$ for the sentence discriminator. For the CSL model, we select the hyperparameter $\alpha = 5, \beta = 4$. During the prediction phase, we decode top-5 answer spans per sentence. Our model is trained using Adamax (Kingma & Ba, 2015) with default hyperparameters.

## 3.3 OVERALL RESULTS

| Datasets | TriviaQA(unfiltered) | | Quasar-T | | SearchQA | |
|----------|------|------|------|------|------|------|
| Models | EM | F1 | EM | F1 | EM | F1 |
| Denoise OpenQA (Lin et al., 2018) | 48.7 | 56.3 | 42.2 | 49.3 | 58.8 | 64.5 |
| Re-Ranker (Wang et al., 2018b) | 50.6 | 57.3 | 42.3 | 49.6 | 57.0 | 63.2 |
| S-Norm (Clark & Gardner, 2018) | 61.6 | 67.6 | - | - | - | - |
| Our OpenQA model + DISTANT* | 57.9 | 63.1 | 61.0 | 65.9 | 58.9 | 64.2 |
| SSL | 61.9 | 66.4 | 61.4 | 66.6 | **59.5** | **65.1** |
| CSL | **63.7** | **68.2** | **62.2** | **67.5** | 59.4 | 64.9 |

Table 2: Experimental results of the OpenQA task on Quasar-T, SearchQA and TriviaQA. We compare our results with three state-of-the-art models: Denoise OpenQA (Lin et al., 2018), Re-Ranker (Wang et al., 2018b) and S-Norm (Clark & Gardner, 2018). *: Our aggregation-based OpenQA model trained only using the distantly supervised label.

The performances of various models on the OpenQA task are shown in Table 2. First, our model without semantic labeler outperforms the re-ranker model (Wang et al., 2018b) and the denoising OpenQA model (Lin et al., 2018) on three datasets by large margins (up to 10-20%). It indicates that by explicitly considering the matches between the question and sentences, our model could more effectively leverage structure information on sentence level to promote the overall performance of the OpenQA task.

Second, we also compare our CSL model with a strong baseline model S-Norm (Clark & Gardner, 2018) and our OpenQA model trained without the semantic labeler on the unfiltered TriviaQA dataset. Our CSL model improves the performance by over 2% EM score as compared to S-Norm and over 5% EM score as compared to our model without the semantic labeler. It demonstrates that our model could achieve state-of-the-art performance on the OpenQA task via utilizing supervised information from the supervised RC dataset.

Finally, we compare the results of our two denoising models. On TriviaQA and Quasar-T, the CSL model achieves consistent improvement against our SSL model. It verifies that through collaborative learning, the CSL model reduces the information gap between the OpenQA dataset and RC dataset, which further improves the OpenQA system performance. Whereas on SearchQA, the SSL model even achieves slightly higher than the CSL model. We examine samples from the three datasets and find that, unlike the previous two datasets, in SearchQA, background sentences are constructed by web snippets that are disparate from paragraphs in SQuAD. This large domain gap inevitably increases the difficulty to learn from both domains and therefore the collaborative learning strategy barely works.

## 3.4 PERFORMANCE OF THE SENTENCE DISCRIMINATOR

| Datasets | TriviaQA(unfiltered) | Quasar-T | SearchQA |
|---|---|---|---|
| Models | Top1 | Top1 | Top1 |
| Paragraph Selector (Lin et al., 2018) | - | 27.7 | 58.9 |
| Semantic Labeler (pretrained on SQuAD) | 38.8 | 34.7 | 52.3 |
| Sentence Discriminator + DISTANT | 54.4 | 59.3 | 71.6 |
| Sentence Discriminator + SEMANTIC | **57.4** | **62.6** | **72.6** |

Table 3: Comparison of our sentence discriminator and other baselines. The DISTANT and SEMANTIC model use distantly supervised labels and semantic labels for training respectively.

For further comparison, we employ two models as our baselines, including the semantic labeler pre-trained on SQuAD and the paragraph selector proposed by Lin et al. (2018). Since distant-supervised sentences are not annotated with accurate labels, we regard a sentence to be the ground truth sentence if it contains the answer span to the question following Lin et al. (2018). Note that this evaluation metric, though not accurate enough, provides a clue on the model's ability to select potential sentences.

Table 3 shows the experimental results. First, both trained with distant supervised labels, our sentence discriminator model still outperforms the paragraph selector significantly. We found that the paragraph selector fails to capture sentence-level feature due to implicitly considering the matches between a sentence and the given question in its optimization goal. Second, our sentence discriminator trained with semantic labels achieves the highest top-1 accuracy compared with models trained only with the supervised source data or the distantly supervised target data. It indicates that our semantic labeler could effectively utilize supervised information to help denoising distant-supervised data.

We evaluate the two types of labels on an example from Quasar-T in Table 4 for the case study. While the first three sentences both contain the correct answer span '*judo*', only the second sentence is correlated with the given question logically. We show that the semantic labeler effectively captures the relatedness between the ground truth sentence and the question by giving a high semantic label to the second sentence. The fourth sentence does not contain the ground truth answer but shares the same key phrase '*gentle way*' with the question. In this case, the semantic labeler fails to compute an accurate label. Therefore, we take advantage of both the distant and semantic labels when training the OpenQA model to benefit from the information in two domains.

| question: *Which sport has a name which literally means 'gentle way'?* | label | |
|---|---|---|
| **Ground truth:** judo | distant | semantic |
| The term "do way", which is used in the names of arts like **judo**, aikido ... | 1 | 0.53 |
| Sport and beyond despite the literal meaning of **judo** being 'gentle way' ... | 1 | 0.94 |
| Kano took the name judo from jikishin ryu **judo**, which is an older school ... | 1 | 0.34 |
| Dr. Kano meant for his gentle way to be a way to live, a path to follow. | 0 | 0.91 |

Table 4: An example on Quasar-T, where sentences are given semantic labels and distantly supervised labels. We evaluate the sentence by whether it is actually supportive to the question. Text span in blue is the ground truth answer for the question, while text spans in red are not.

### 3.5 PERFORMANCE WITH DIFFERENT PROPORTION OF SUPERVISED DATA

To demonstrate the robustness of our denoising strategies, we also study the denoising performance when the model is exposed to a different proportion of the supervised data on unfiltered TriviaQA. As shown in Figure 3, the performances of both models degrade when less supervised data is used. Nevertheless, the collaborative model is more robust comparing to the semi-supervised model when only a minority of supervised RC data is provided.

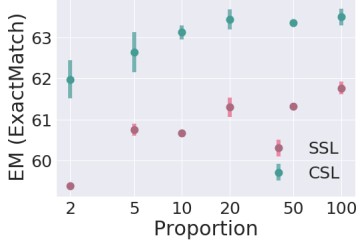 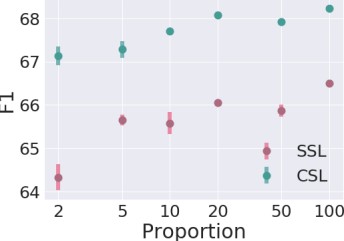

Figure 3: Experimental results on unfiltered TriviaQA shows denoising performance varies with different porportion of supervised data.

## 4 RELATED WORKS

**Open-domain Question Answering.** The task of open-domain question answering was first proposed by Green Jr et al. (1961). It aims to answer the given question by exploiting external resources such as documents (Voorhees et al., 1999), webpages (Kwok et al., 2001) and structured knowledge graphs (Berant et al., 2013; Bordes et al., 2015). Previous efforts (Chen et al., 2017; Dhingra et al., 2017a; Cui et al., 2017) follow a two-step procedure, i.e., they first retrieve a collection of passages using an information retrieval system and then apply a reading comprehension model to extract the answer. These models usually fail to handle the noise in the first place, where no passage-level annotations are provided. Thus, recent approaches have taken various attempts to filter out noisy contents. Wang et al. (2018b) propose to rerank candidates by aggregating evidence in multiple passages; Lin et al. (2018) and Clark & Gardner (2018) both take all the passages into account and propose techniques to rerank the passages; Min et al. (2018) focus more on selecting a minimal set of sentences with sufficient information. To the best of our knowledge, our work is the first that incorporates supervised information to eliminate noise in the open-domain QA task.

**Semi-supervised learning in NLP.** Semi-supervised learning is widely used in NLP tasks such as dependency parsing, sentimental analysis, named entity recognition, question answering, etc, where human annotations for large-scale datasets are expensive. Corro & Titov (2018) propose a latent-variable generative network to model unlabeled text for semi-supervised parsing. Gupta et al. (2018) apply manifold regularization technique combining the external and in-domain data to improve the performance of low resource sentimental analysis. Xu et al. (2018) introduce a unified model for Chinese named entity recognition combining cross-domain learning and in-domain semi-supervised learning. Yang et al. (2017) propose to train a generative model to obtain question-answer pairs from unlabeled text together with domain adaptation techniques for semi-supervised question answering.

## 5 CONCLUSION

In this paper, we propose an aggregation-based OpenQA model with denoising strategies to leverage the alignment information from supervised RC datasets to handle the noise in the open-domain QA task. Experimental results on three large-scale OpenQA datasets including Quasar-T, TriviaQA, and SearchQA show that our model achieves state-of-the-art results on the OpenQA task. In future work, we will explore our model in exploiting other forms of supervised signals such as external knowledge bases, and we believe our model could be potentially generalized to other types of external information as well.

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
