# OpenReview forum: "Denoise while Aggregating: Collaborative Learning in Open-Domain Question Answering"
_ICLR.cc/2019/Conference_

### Official Review · AnonReviewer3 · 2018-10-21
**Interesting exploration, but weak novelty**

**Rating:** 5
**Confidence:** 4

**Review:**

This paper aims for open-domain question answering with distant supervision. First, the authors proposed an aggregation-based openQA model with sentence discriminator and sentence reader. Second, they use a semantic labeler to handle distant supervision problem by utilizing other span supervision tasks, and propose two different denoising methods. They run experiments on 3 open-domain QA datasets and achieve SOTA.


Strengths

1) Their semantic labeler and exploration of two different denoising methods are interesting and meaningful.
2) They conducted experiments on 3 widely-used open-domain datasets, and the performance gain is impressive.


Weakness

Although there is an impressive performance gain, the contribution of the paper seems to be marginal.
1) First of all, it is hard to say there is a contribution to the idea of sentence discriminator and sentence reader — people have used this framework for large-scale QA a lot. Also, the architecture of the models in this paper are almost identical to Chen et al (ACL 2017) and Lin et al (ACL 2018).
2) Thus, the contribution is more on semantic labeler and denoising method. However, this contribution is marginal as well since its role is almost the same as sentence discriminator plus pretraining methods which have widely used already.


Questions

1) What exactly is the difference between semantic labeler and sentence discriminator? For me, it seems like both of them label each sentence `yes` or `no`. My thought is sentence discriminator is only trained on the target dataset (distant supervision dataset) while semantic labeler is also trained (either jointly or separately) trained on the source dataset (span supervision dataset). (If my thought is wrong, please let me know, I would like to update my score.)
2) Chen et al (ACL 2017) have shown that pretraining QA model on span supervision dataset (SQuAD) is effective to train the model on distant supervision dataset. Similarly, Min et al (ACL 2018) have pretrained both QA model and sentence selector on SQuAD. While I think pretraining sentence selector on SQuAD is almost identical to sentence labeler with SSL method, could you give exact comparison of these different methods? For example, remove sentence labeler, and pretrain both sentence discriminator and reader on SQuAD, or jointly train them on SQuAD & target dataset.


Marginal comments

1) At the beginning of Section 2.4.1, it says the semantic labeler is able to transfer knowledge from the span supervised data — however, the authors should be careful since people usually refers to `knowledge` as an external knowledge. This method is more like better learning of accurate sentence selection, not transferring knowledge.
2) Please mention the TriviaQA data you used is Wikipedia domain, since there are two different domains (Wikipedia and Web).
3) In References section, the conference venues in many papers are omitted.


Overall comments

The paper explored several different methods to deal with distant supervision via sentence labeling, and I really appreciate their efforts. While the result is impressive, the idea in the paper is similar to the methods that have widely used already.

---

### Official Review · AnonReviewer2 · 2018-10-31
**Good results, but the contribution to "denoising" does not hold given the current version**

**Rating:** 6
**Confidence:** 4

**Review:**

This paper proposes a new open-domain QA system which gives state-of-the-art performance on multiple datasets, with a large improvement on QuasarT and TriviaQA. Given the significant results, I would vote for its acceptance.

The contributions of the paper can be summarized into two parts. The first is an efficient base open-domain QA model; and the second includes several denoising methods of passage selection. I hope the authors could address the following issues during rebuttal, which I believe will make the paper stronger.

(1) The proposed base open-domain QA method (+DISTANT) itself improves a lot, which I think is the major contribution of the paper. It will be very helpful if the authors could provide ablation test to the sentence discriminator/reader modules to give better clues about why it works so well. Is it mainly because of the usage of DrQA style encoder?

(2) Although the paper has "denoising" emphasized in the title, I actually do not see this holds as a contribution. First, the proposed semantic labeler training strategies only improve significantly on TriviaQA. The improvement on Quasar-T and SearchQA is relatively marginal. I am wondering whether this is because the domain shift between SQuAD and QuasarT and SearchQA.

(3) (Cont'd from point 2) Second, the proposed SSL and CSL are not compared with any decent baselines. It will make more sense if the authors apply the Re-Ranker or S-Norm to the proposed base open-domain QA model, and compare the improvement from different methods (Re-Ranker, S-Norm, SSL, CSL). It is likely that the S-Norm could also improve on TriviaQA and the Re-Ranker could improve over Quasar-T and SearchQA. Therefore without any experimental evidence, again, the "denoising" part cannot be regarded as a contribution with positive results.

Additional question: would the authors release the code for public usage if the paper gets accepted?

---

### Official Review · AnonReviewer1 · 2018-11-02
**Pre-training for QA helps**

**Rating:** 4
**Confidence:** 4

**Review:**

This paper shows that a sentence selection / evidence scoring model for QA trained on SQuAD helps for QA datasets where such explicit per-evidence annotation is not available.

Quality:
Pros: The paper is mostly well-written, and suggested models are sensible. Comparisons to the state of the art are appropriate, as is the related work description. the authors perform a sensible error analysis and ablation study. They further show that their suggested model outperforms existing models on three datasets.
Cons: The introduction and abstract over-sell the contribution of the paper. They make it sound like the authors introduce a new task and dataset for evidence scoring, but instead, they merely train on SQuAD with existing annotations. References in the method section could be added to compare how the proposed model relates to existing QA models. The multi-task "requirement" is implemented as merely a sharing of QA datasets' vocabularies, where much more involved MTL methods exist. What the authors refer to as "semi-supervised learning" is in fact transfer learning, unless I misunderstood something.

Clarity:
Apart from the slightly confusing introduction (see above), the paper is written clearly.

Originality:
Pros: The suggested model outperforms others on three QA datasets.
Cons: The way achievements are achieved is largely by using more data, making the comparison somewhat unfair. None of the suggested models are novel in themselves. The evidence scoring model is a rather straight-forward improvement that others could have come up with as well, but merely haven't tested for this particular task.

Significance:
Other researchers within the QA community might cite this paper and build on the results. The significance of this paper to a larger representation learning audience is rather small.

---

### Meta-Review · Area_Chair1 · 2018-12-14

**Confidence:** 4
**Recommendation:** Reject

**Metareview:**

This paper presents a model for question answering, where the idea is to have a collaborative model that aligns queries and sentences on a small supervised dataset and also uses semi-supervised information from a weakly supervised corpus to answer open domain questions resulting in short answer spans.

The main criticism of the paper is regarding its novelty, and reviewers cite the similarities with prior work such as Chen et al. and Min et al.  There is relative consensus between the reviewers that further work using the semi-supervised outlook with stronger results could strengthen the paper further.